

# Non-stabilizerness of Sachdev-Ye-Kitaev model

**Surajit Bera⋆ and Marco Schirò**

JEIP, UAR 3573 CNRS, Collège de France, PSL Research University,
11 Place Marcelin Berthelot, 75321 Paris Cedex 05, France

⋆ surajit.bera@college-de-france.fr

## Abstract

We study the non-stabilizerness or quantum magic of the Sachdev-Ye-Kitaev (SYK) model, a prototype example of maximally chaotic quantum matter. We show that the Majorana spectrum of its ground state, encoding the spreading of the state in the Majorana basis, displays a Gaussian distribution as expected for chaotic quantum many-body systems. We compare our results with the case of the $SYK_2$ model, describing non-chaotic random free fermions, and show that the Majorana spectrum is qualitatively different in the two cases, featuring an exponential Laplace distribution for the $SYK_2$ model rather than a Gaussian. From the spectrum we extract the Stabilizer Renyi Entropy (SRE) and show that for both models it displays a linear scaling with system size, with a prefactor that is larger for the SYK model, which has therefore higher magic. Finally, we discuss the spreading of quantun magic under unitary dynamics, as described by the evolution of the Majorana spectrum and the Stabilizer Renyi Entropy starting from a stabilizer state. We show that the SRE for the $SYK_2$ model equilibrates rapidly, but that in the steady-state the interacting chaotic SYK model has more magic than the simple $SYK_2$. Our results suggest that the Majorana spectrum is qualitatively distinct in chaotic and non-chaotic many-body systems.

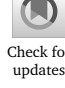

# 1  Introduction

In the last decades concepts and ideas from quantum information and computation have found rich and fruitful applications in quantum many-body physics. One prominent example is quantum entanglement which is both a fundamental resource for many quantum information tasks [1] and a powerful tool to characterize the different phases of matter and their behavior in and out of equilibrium [2–4]. Entanglement is however only one of the quantum resources that are necessary for advantage in performing certain quantum tasks [5]. In particular, it is known that entanglement by itself is not sufficient to achieve quantum computational advantage over classical simulations. This is clear since there exist quantum circuits based on the Clifford group and the stabilizer formalism that can generate extensive entanglement yet can be simulated in polynomial time [6–8]. While Clifford resources are free, they are not sufficient to ensure universal quantum computation. Non-stabilizer or magic states are in this sense expensive yet crucial resources to achieve possible advantage [9, 10].

As the effort and interest around the goal of building a universal quantum computer become more concrete, the task of simulating quantum many-body systems emerge as a prominent source of potential advantage. It has therefore become urgent to measure the degreee of non-stabilizerness or quantum magic of quantum many-body states [11] and to understand its connection to other characteristic features of quantum dynamics, such as chaos and scrambling [12]. This has sparked interest towards new, scalable measures of quantum magic [13–16] among which the Stabilizer Renyi Entropy [17, 18] has received particular attention. A number of theoretical works have recently explored the quantum magic structure of different many-body systems, including spin chains [19–26], field theories and lattice gauge theories [27–29], matrix product states [30–33], random circuits [34–40], fermionic gaussian states [41].

One intriguing aspect of quantum magic is that it is crucial to obtain a non-trivial quantum chaos structure, which is lacking in Clifford based resources [42]. It is therefore particularly interesting to study quantum magic and non-stabilizerness of chaotic quantum many-body systems. A toy model for quantum many-body chaos which has recently attracted large interest across communities is the Sachdev-Ye-Kitaev model [43–45], describing fermions with random all to all interactions. A notable feature of the SYK model is that it is the simplest toy model

to describe quantum matter without quasiparticles, such as Non-Fermi-Liquids [46] or Black Holes [43, 47–49], and that it is maximally chaotic in the sense that it saturates the quantum bound on chaos [44,50]. Furthermore it is known to possess entangled many-body ground and excited states possessing volume law scaling [51]. The SYK model has also attracted interest at the interface between holography, quantum matter and quantum error corrections [52–55]

With these motivations in this work we study the non-stabilizerness of the SYK model, both in its groundstate as well as in a state generated by unitary quantum dynamics starting from a stabilizer state. In particular we compute the Majorana spectrum, which encodes the spreading of a quantum state in the Majorana basis and from which we obtain the stabilizer Renyi Entropy. We compare the results for the SYK model with a model of random Gaussian fermions, the so called SYK$_2$. Our results show that the Majorana spectrum of the two models differ qualitatively, both in the ground-state and in the dynamics: the maximally chaotic SYK model shows a Gaussian distribution of Majorana strings - in agreement with recent results on random circuits and chaotic Hamiltonian, while the SYK$_2$ has broader distribution compatible with an exponential one. The corresponding SRE displays similar qualitative scaling with system size, yet the SYK is found to display more magic than the random free fermion model.

The manuscript is structured as follows. In Sec. 2 we set the stage and review the stabilizer formalism for fermions and the definition of stabilizer Renyi entropy that will be used throughout this work. We introduce the SYK model and recall briefly its properties, together with the random fermion model in Sec. 3. In Sec. 4 we discuss the methods used to compute the SRE, in particular the Monte Carlo sampling of Majorana strings. Finally in Sec. 5 we present our results on quantum magic and the Majorana spectrum of the SYK ground-state, as well as the dynamics of magic. Finally, in Sec. 6 we summarize our results and draw our conclusions and discuss potential future research directions.

## 2 Preliminaries and definitions

The stabilizer formalism for qubits is typically introduced by first defining the Pauli group, which consists of all Pauli string operators, followed by the definition of operations that leave Pauli strings invariant, known as the Clifford group [6,8]. By applying Clifford operations, one can realize error-correcting codes such as Toric codes, which serve as a fundamental building block of fault-tolerant quantum computation. The universality of quantum computation is achieved through the inclusion of non-Clifford operations, which are the key resource for exponential quantum advantage in computation. The stabilizer formalism and the associated measure of non-stabilizerness have been extensively studied in recent times, particularly in the context of spin systems. For fermionic systems, an analogous stabilizer formalism can be formulated by employing Majorana fermions [56].

To introduce the concept of Clifford group and stabilizer states for fermions, we follow recent results [41,57–59] and introduce first the concept of Majorana strings and the Majorana group, which generalizes the Pauli group, and that it is in one-to-one correspondence with Pauli group [59]. To this end, we consider a set of $N$ complex fermionic degrees of freedom, $c_i$ and $c_j^\dagger$, satisfying the algebra

$$\{c_i, c_j\} = 0, \qquad \{c_i^\dagger, c_j^\dagger\} = 0, \qquad \{c_i, c_j^\dagger\} = \delta_{ij}. \tag{1}$$

It is convenient to introduce $2N$ Majorana degrees of freedom, $\eta_i$ and $\chi_i$ $(i = 1, \ldots, N)$, associated with these fermions, defined as

$$\eta_i = c_i + c_i^\dagger, \tag{2}$$

$$\chi_i = \mathrm{i}\left(c_i - c_i^\dagger\right), \tag{3}$$

with the following properties: $\{\eta_i, \eta_j\} = 2\delta_{ij} = \{\chi_i, \chi_j\}$ and $\eta_i^2 = \chi_i^2 = 1, \{\eta_i, \chi_j\} = 0$. One can then introduce a Hermitian Majorana string of length $2N$ [59], defined as

$$\hat{\mu}(v) = (\mathrm{i})^{v^T \omega_L v} \cdot \eta_1^{v_1} \chi_1^{v_2} \cdots \chi_N^{v_{2N}}, \tag{4}$$

where $v^T = \begin{pmatrix} v_1 & v_2 & \dots & v_{2N} \end{pmatrix}$ is a $2N$-dimensional vector, and $v_i \in \mathbb{Z}_2$ ($i = 1, \dots, 2N$) takes values 0 or 1, specifying which Majorana operators are present in the Hermitian Majorana string. The factor $(\mathrm{i})^{v^T \omega_L v}$ [59], which ensures the Hermiticity of Majorana strings, yields real expectation values—an essential requirement for introducing the Majorana spectrum, as discussed later. Here, $\omega_L$ is a lower triangular matrix where all elements below the diagonal are 1, while all diagonal and above-diagonal elements are 0. The matrix product $v^T \omega_L v \in \mathbb{Z}_4$. In simple terms, when the Majorana string is anti-Hermitian, it is multiplied by $\pm \mathrm{i}$ to make it Hermitian.

The Majorana strings have a one-to-one correspondence with the Pauli string operators by employing the Jordan-Wigner transformation. The set of all Hermitian Majorana strings of length $2N$ forms a non-abelian Majorana group $\mathcal{M}_{2N}$ under operator multiplication. The total number of elements in $\mathcal{M}_{2N}$ is $2^{2N}$. The Majorana Clifford group $\mathcal{C}_{2N}$ is then defined as the group of all unitary operators $\mathcal{U}$ that, when acting on a Majorana string in $\mathcal{M}_{2N}$, maps it to another Majorana string under conjugation, i.e., $\mathcal{U}\hat{\mu}(v)\mathcal{U}^\dagger \equiv \hat{\mu}(v')$. For fermionic systems, due to the parity superselection rule, all the operators for physical observables must commute with the parity operator $\hat{\mathcal{P}} = (-\mathrm{i})^N \gamma_1 \gamma_2 \cdots \gamma_{2N} = Z_1 \cdots Z_N$, where $Z$ denotes the Pauli-$Z$ operator. The Majorana string with only even-parity commutes with the parity operator $\hat{\mathcal{P}}$, and can describe a physical observable or stabilizer operation. On the other hand, odd-parity strings can act as logical operations. The set of even-parity Majorana strings that obey abelian operations forms stabilizer group which is an abelian subgroup within Majorana group. The stabilizer states in this context are those pure quantum states that can be prepared by means of Clifford operations only, starting from the initial trivial state $|0\rangle^{\otimes N}$.

## 2.1 Majorana spectrum and stabilizer Rényi entropy

The non-stabilizerness or magic of a quantum state is the measure that quantifies how a quantum state differs from a stabilizer state. Among the various measures of non-stabilizerness of a quantum state, the decomposition of a quantum state in a complete operator basis in the Hilbert space, and the associated calculation of the Rényi entropy of the resulting probability distribution, serves as an efficient tool for studying magic [17]. This approach avoids computationally expensive and inefficient minimization procedures.

In the case of fermionic systems, the operators $\hat{\mu}(v)$ in the Majorana group $\mathcal{M}_{2N}$ satisfy the completeness relation and orthonormal condition given by: $\mathrm{Tr}(\hat{\mu}(v)\hat{\mu}(v')) = d\,\delta_{v,v'}$, where $d = 2^N$ is the Hilbert space dimension of the quantum state for $N$ complex ($2N$-Majorana) fermions. Thus, a quantum state $\rho = |\Psi\rangle\langle\Psi|$ can be decomposed in the Majorana operator basis as:

$$\rho = |\Psi\rangle\langle\Psi| = \frac{1}{d} \sum_v \langle\Psi|\hat{\mu}(v)|\Psi\rangle \hat{\mu}(v). \tag{5}$$

The amplitude of the quantum state, $\langle\Psi|\hat{\mu}(v)|\Psi\rangle$, can be referred to as the Majorana spectrum in this decomposition, drawing an analogy with the Pauli spectrum for qubits [34]. From the condition $\mathrm{Tr}(\rho^2) = 1$ for a pure state, one obtains: $\frac{1}{d} \sum_v \langle\Psi|\hat{\mu}(v)|\Psi\rangle^2 = 1$. This allows the Majorana spectrum to be interpreted as a probability distribution. We define the following distribution function:

$$\Pi(x) = \frac{1}{d^2} \sum_{y \in \{\langle\Psi|\hat{\mu}(v)|\Psi\rangle\}} \delta(x - y). \tag{6}$$

The $\alpha$-moments of this distribution are given by:

$$\zeta_\alpha = d \int dx\, \Pi(x) x^{2\alpha} = \sum_{\hat{\mu}(\nu)} \frac{\langle\Psi|\hat{\mu}(\nu)|\Psi\rangle^{2\alpha}}{d}\,. \tag{7}$$

The $\alpha$-stabilizer Rényi entropy (SRE) for the probability distribution of the quantum state decomposition in the Majorana operator basis is related to the moments $\zeta_\alpha$ via the following relation:

$$M_\alpha = \frac{1}{1-\alpha} \log_e[\zeta_\alpha]\,. \tag{8}$$

For $\alpha \to 1$, the above expression reduces to Shannon entropy of the probability distribution $\langle\Psi|\hat{\mu}(\nu)|\Psi\rangle^2/d$. The moments $\zeta_\alpha$ can be interpreted as the inverse participation ratio, and the SRE as the participation entropy [34] in the complete Majorana operator basis.

In addition to the SRE, it has been found that the filtered version of the SRE [34] is highly effective in distinguishing between typical and atypical quantum states in the Hilbert space. The filtered SRE for fermionic systems [41] removes the contributions from the identity operator $\hat{I} = \mathcal{I}^{\otimes N}$ and the parity operator $\hat{\mathcal{P}}$ as $\langle\Psi|\hat{\mathcal{I}}|\Psi\rangle^2 = 1 = \langle\Psi|\hat{\mathcal{P}}|\Psi\rangle^2$ for pure fermionic state $|\Psi\rangle$. The filtered SRE is defined as follows:

$$\tilde{M}_\alpha = \frac{1}{1-\alpha} \log\left[ \sum_{\hat{\mu}(\nu)\notin\{\hat{I},\hat{\mathcal{P}}\}} \frac{\langle\Psi|\hat{\mu}(\nu)|\Psi\rangle^{2\alpha}}{d-2} \right], \tag{9}$$

where the denominator $d-2$ ensures that stabilizer states have zero filtered SRE. This filtering is particularly useful for large system sizes $N$ and for $\alpha > 2$, where the dominance of the contributions from $\hat{I}$ and $\hat{\mathcal{P}}$ becomes significant.

## 3 Sachdev-Ye-Kitaev model and its variants

In this work, we consider models of complex fermionic degrees of freedom interacting via all-to-all random couplings. The *zero*-dimensional SYK model falls into this class and it is described by the following Hamiltonian:

$$H_{\text{SYK}} = \sum_{ijkl} J_{ijkl} c_i^\dagger c_j^\dagger c_k c_l - \mu \sum_i c_i^\dagger c_i\,, \tag{10}$$

where the couplings $J_{ijkl}$ are drawn from an anti-symmetrized complex random Gaussian distribution with variance $\langle J_{ijkl}^2 \rangle = J^2/(2N)^3$. Here, $i, j, k, l = 1, \ldots, N$, with $N$ representing the total number of fermionic sites or flavors, and $\mu$ is the chemical potential. The SYK model is exactly solvable in the large-$N$ limit and describes non-Fermi liquid ground state without quasi-particles. Furthermore, the SYK model exhibits peculiar zero-temperature residual entropy, acts as a fast scrambler by saturating the bound of quantum chaos, and can be used to describe certain black holes via holographic correspondence. In this work, we focus on the non-stabilizer content of many-body quantum states in the SYK model for finite-size systems.

To facilitate the comparison and understanding of the 'magic' structure of the SYK model, we will also consider a non-interacting variant of the SYK model, which describes Fermi liquid ground state, referred to as the SYK$_2$ model. Its Hamiltonian is given by:

$$H_{\text{SYK}_2} = \sum_{ij} J_{ij} c_i^\dagger c_j - \mu \sum_i c_i^\dagger c_i\,, \tag{11}$$

where $J_{ij}$ are complex random Gaussian numbers with variance $J^2/N$. This model does not exhibit many-body chaotic behavior, as it is a non-interacting model. The quantum information theoretic quantity entanglement for the ground states of both models has been studied

earlier [60–64] (see Ref. [51] for a recent review), revealing that they follow a volume-law scaling, as both models describe all-to-all couplings. In the following, we will discuss how the resource theoretic quantity 'magic' of the many-body quantum states of these models compares, and whether the concept of 'magic' can distinguish between their many-body states. To this end, we employ exact diagonalisation (ED) and Monte Carlo method integrated with ED to study the 'magic' of both models as we discuss below. In all of the following, we set $J = 1$ as unit of energy for both $SYK_2$ and SYK models. Throughout this work, we consider zero chemical potential ($\mu = 0$) and perform exact diagonalisation within the half-filling sector.

## 4 Exact diagonalization and Monte Carlo sampling

The numerical evaluation of the stabilizer entropy or the full Majorana spectrum is challenging in the many-body context for two primary reasons. First, one requires access to the full many-body wavefunction $|\Psi\rangle$. Second, one must compute an exponentially large number of expectation values of Majorana string operators, namely $d^2 = 2^{2N} = 4^N$.

For exact diagonalization, we write down the Hamiltonian $H$ in the occupation basis $|n\rangle = |n_1 n_2 \ldots n_N\rangle$, where $n_i = c_i^\dagger c_i$ ($i = 1, \ldots, N$) represents the occupation number of the $i$-th site. We numerically diagonalize the Hamiltonian to obtain its eigenvalues and eigenstates. The total particle number operator $\mathcal{N} = \sum_i n_i$ is a conserved quantity, as it commutes with the Hamiltonian. Therefore, we can directly work within a fixed particle number sector. The many-body wavefunction in the full Fock space basis has the following structure:

$$|\Psi\rangle = \bigoplus_{\mathcal{N}=0}^{N} |\Psi_{\mathcal{N}}\rangle, \tag{12}$$

where $|\Psi_{\mathcal{N}}\rangle$ represents the many-body state for a fixed particle number $\mathcal{N} = N_p$. In this work, we consider the many-body quantum state with a fixed particle number $N_p = N/2$. Consequently, in the quantum state $|\Psi\rangle$, we only have a non-trivial wavefunction in the $N_p = N/2$ sector, while all other sectors have zero contributions due to the fermionic parity superselection rule. Once the many-body quantum state $|\Psi\rangle$ is obtained, we can, in principle, compute all the expectation values of the Majorana string operators for any system size accessible via ED.

Due to fermionic parity and superselection rules, exactly $d^2/2$ expectation values of Majorana operators with odd parity (where the total number of Majorana operators in a string is odd) are *zero*. Thus, we only need to compute the remaining $d^2/2$ expectation values for the even-parity Majorana string operators. While this reduces the computational cost, the task remains exponentially large in $N$. Although exact diagonalisation (ED) for all-to-all models such as the SYK model is feasible up to system sizes $N = 16$, the exact enumeration of all $d^2/2 = 4^N/2$ Majorana string expectation values becomes computationally tractable only up to $N \leq 8$, particularly when averaging over multiple disorder realizations. For intermediate system sizes $N = 10, 12, 14$, we avoid exact enumeration of all expectation values due to the numerical expense. Instead, we distill important samples from the $d^2/2$ even-parity Majorana string operators. Below, we discuss the Monte Carlo sampling method.

### 4.1 Monte Carlo sampling

For distilling the important sampling, we use the Metropolis-Hastings Markov chain sampling method, similar to the Pauli string sampling introduced for spin systems in Ref. [29], generalized for Majorana string operators. We sample the contributions of the expectation values of

string operators according to the distribution

$$\sigma_v = d^{-1} \langle \Psi | \hat{\mu}(v) | \Psi \rangle^2 . \tag{13}$$

For the above probability distribution, the stabilizer Rényi entropy for $\alpha \geq 2$ is obtained using the expression

$$M_\alpha = \frac{1}{1-\alpha} \log \left\langle \langle \Psi | \hat{\mu}(v) | \Psi \rangle^{2(\alpha-1)} \right\rangle_{\sigma_v}, \tag{14}$$

and for $\alpha = 1$,

$$M_1 = - \left\langle \log \langle \Psi | \hat{\mu}(v) | \Psi \rangle \right\rangle_{\sigma_v}, \tag{15}$$

where $\left\langle \cdots \right\rangle_{\sigma_v}$ represents the average over all sampled contributions obtained with probability distribution $\sigma_v$. From the above, we define the estimator for the sampling procedure as

$$X_v = \begin{cases} \log \langle \Psi | \hat{\mu}(v) | \Psi \rangle, & \text{for } \alpha = 1, \\ \langle \Psi | \hat{\mu}(v) | \Psi \rangle^{2(\alpha-1)}, & \text{for } \alpha \geq 2. \end{cases} \tag{16}$$

To obtain the stabilizer Rényi entropy (SRE), we propose updates involving Majorana operators of two sites from the set $\{\gamma_i, \eta_i\}$ to construct a new Majorana string of size $2N$, starting from an initial even number of Majorana string operator. This is done under the constraint that the total number of Majorana operators, $\gamma_i$ and $\eta_i$, remains even due to the fermionic parity superselection rule. The detailed implementation of the two site updates is discussed in Appendix B. The algorithm for our Monte Carlo sampling is given in Algorithm 1.

The calculation of the SRE via direct sampling becomes inefficient for large systems, as a significant fraction of the sampling steps are dominated by contributions from the identity operator $\mathcal{I}$ and the parity operator $\mathcal{P}$. In contrast, in the computation of the filtered SRE, these contributions are excluded by setting the probabilities $\sigma_v[\mathcal{I}, \mathcal{P}] = 0$. In Appendix C, we show the convergence of the filtered SRE (Figure 7) and direct sampling of SRE (Figure 8) results with respect to the sampling size $N_S$ and benchmark these results against exact diagonalisation (ED) for smaller system sizes up to $N = 12$ for a particular disorder realisation. We find that the convergence of the filtered SRE with sampling size is relatively better than that of the direct sampling approach. Furthermore, we observe that the Monte Carlo convergence of filtered SRE for the SYK model appears to be relatively better than for the SYK$_2$ model. Based on these observations for convergence with $N_S$, we set the sampling size to $N_S = 5 \times 10^5$ for system sizes $N = 10, 12, 14$ for both the filtered SRE and the direct SRE results presented in the main text. All results for the ground state filtered SRE presented in the main text are computed using filtered SRE sampling, while results for the time evolution are computed using direct SRE sampling method.

## 5 Results

### 5.1 Non-stabilizerness of the ground state

Here, we discuss the results obtained using ED and Monte Carlo sampling for non-stabilizerness of the many-body ground state at half-filling ($N_p = N/2$) for SYK$_2$ and SYK, characterized by the Majorana spectrum and Stabilizer Renyi entropy (SRE).

---

**Algorithm 1** Monte Carlo Sampling of Majorana String

---

1: **procedure** MARKOVSAMPLING($\Psi, N_S$)
2:     Initialize *Majorana string operator* $\hat{\mu}(v)$
3:     Compute $\sigma_v = d^{-1}\langle\Psi|\hat{\mu}(v)|\Psi\rangle^2$, and the estimator $X_v$
4:     **for** $i = 1$ to $N_S$ **do**
5:         Propose a new *Majorana string operator* $\hat{\mu}(v')$
6:         Compute $\sigma_{v'} = d^{-1}\langle\Psi|\hat{\mu}(v')|\Psi\rangle^2$
7:         Accept the move with probability: $\min\left(1, \frac{\sigma_{v'}}{\sigma_v}\right)$
8:         **if** the move is accepted **then**
9:             Measure the estimator $X_{v'}$
10:         **end if**
11:     **end for**
12:     **return** list of Majorana strings $\{\hat{\mu}(v)\}$, and list of estimators $\{X_v\}$
13: **end procedure**

---

### 5.1.1 Majorana spectrum for the ground state

In this section, we discuss the Majorana spectrum for the SYK and SYK$_2$ models. We find that the spectrum encodes valuable information about the chaotic or non-chaotic nature of the many-body Hamiltonian.

In Fig. 1, we show the Majorana spectrum $\Pi(x)$ for the ground state at half-filling for both models with system size $N = 12$, obtained using the exact diagonalisation (ED) method. These distributions are plotted for a single disorder realization of the random couplings. As we detail in Appendix A, the Majorana spectrum displays weak sample-to-sample fluctuations and behaves as a self-averaging quantity. Furthermore, we show the Majorana spectrum for an ensemble of different disorder realizations for different system sizes. In all the Majorana spectra shown in this work, we have removed trivial *zero* values of $d^2/2$ expectation values of odd parity Majorana strings, which would otherwise give a delta function peak at $x = 0$. Additionally, we also remove the distinct peak at $x = 1$, arising from the contributions of the identity operator ($\hat{I}$) and the parity operator ($\hat{\mathcal{P}}$) in the Majorana spectrum as we discuss below.

In Fig. 1(a), the distribution of $\Pi(x)$ is presented in semi-logarithmic ($\log y$) scale for the SYK$_2$ model. In Fig. 1(b), the distribution of $\Pi(x)$ is presented in semi-logarithmic ($\log y$) scale for the SYK model. We observe that the distribution of the SYK$_2$ model exhibits a sharp peak at $x = 0$ and a broad tail, while the distribution for the SYK model has a dome-like structure and is relatively narrower compared to the SYK$_2$ model.

We now characterize more precisely the functional form of the connected part of Majorana spectrum, starting from the SYK model. Previous work on random circuits and chaotic many-body systems has shown that the Pauli spectrum follows a Gaussian distribution, as expected from quantum typicality arguments [34]. Since the SYK model is chaotic, as known by the level statistics and the volume-law entanglement of its eigenstates including its ground-state [51], we parametrize the Majorana spectrum following Ref. [34] as

$$\Pi(x) = \left(\frac{d^2 - 4}{2d^2}\right)\frac{e^{-\frac{x^2}{2b}}}{\sqrt{2\pi b}} + \frac{2}{d^2}\delta(x-1) + \frac{D_0}{d^2}\delta(x), \tag{17}$$

where the Dirac-delta at $x = 1$ accounts for the parity and identity operator while the one at $x = 0$ for the $D_0 = d^2/2$ strings with odd numbers of Majorana fields. The variance of this distribution, $b$, can be fixed by imposing the constraint [34] $\int dx\, x^2 \Pi(x) = 1/d$. The above expression perfectly fit the data for what concerns the connected component, see Fig. 1(b).

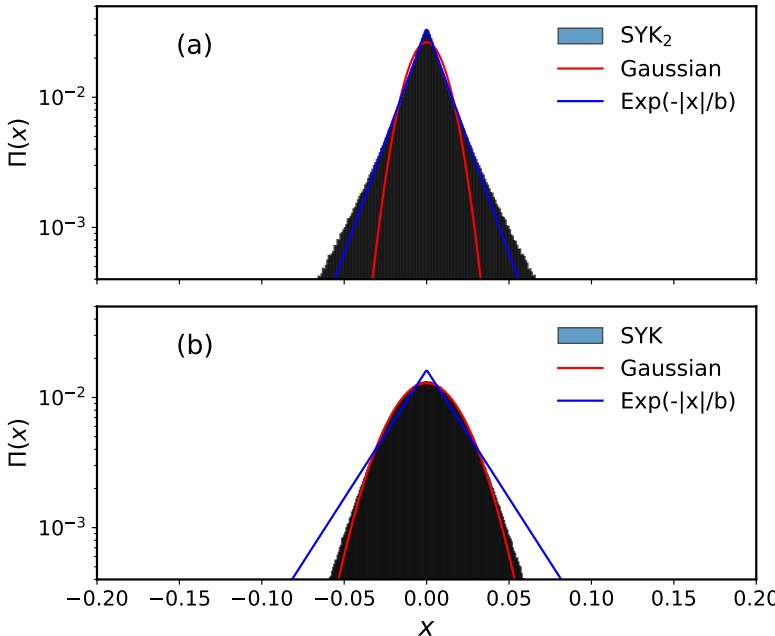

Figure 1: (a) The Majorana spectrum $\Pi(x)$ for the ground state of SYK$_2$ model as a function of $x = \langle \Psi | \hat{\mu}(\nu) | \Psi \rangle$ is shown as a histogram in semi-logarithmic ($\log y$) scale. The spectrum clearly follows a Laplace distribution $\sim \exp(-|x|/b)$. (b) The Majorana spectrum $\Pi(x)$ for the ground state of SYK model as a function of $x$ is shown as a histogram in semi-logarithmic ($\log y$) scale. The spectrum clearly follows a Gaussian distribution. Both (a, b) are computed from system size $N = 12$ at half-filling.

.

This is particularly true at small values of $x$, while the tails of the distribution appears to be exponentially suppressed. Such a feature was found also in non-integrable spin chains [34], and is a generic feature of chaotic many-body system. The structure of these tails and their content in terms of Majorana strings is an interesting question that we leave for the future. We limit to note that Ref. [34] attributes the tails to (exponentially rare) local Pauli strings which thermalize in agreement with the Eigenstate Thermalization Hypothesis.

We now move to the SYK$_2$ case which we show, quite interestingly, to display a qualitatively different distribution of the Majorana spectrum. Indeed, as we show in Fig. 1 (a), in this case $\Pi(x)$ is non Gaussian: in the semi-logarithmic scale, the distribution in Fig. 1(a) clearly shows a $-|x|$ form. We model the numerical results by fitting $\Pi(x)$ with a Laplace distribution, i.e.

$$\Pi(x) = \left( \frac{d^2 - D_0 - 2}{d^2} \right) \frac{e^{-|x|/b}}{2b} + \frac{2}{d^2} \delta(x-1) + \frac{D_0}{d^2} \delta(x), \qquad (18)$$

where again we have included two Dirac-delta contributions at $x = 1$ and $x = 0$. We note that for free fermions the number vanishing Majorana strings is higher than $d^2/2$ and it is given by $D_0 = d^2 - \binom{2N}{N}$, as discussed in Ref. [19]. Again, we can fix the coefficient $b$ in the Majorana spectrum by imposing the same constraint as before on the spectrum, $\int dx\, x^2 \Pi(x) = 1/d$, which gives

$$b^2 = \frac{d-2}{2(d^2 - D_0 - 2)} \,. \qquad (19)$$

We note that the scaling of this coefficient for large $N$ as $b^2 \sim 2^{-N}\sqrt{N}$ gives rise to logarithmic corrections to the SRE, as discussed in Ref. [41]. As shown in Fig. 1(a), the Laplace distribution fits the data very well, confirming that the distribution deviates significantly from a Gaussian form, also shown for comparison in the plot. We emphasize that Eq. (18) represent a purely phenomenological ansatz for the Majorana spectrum of the $SYK_2$ model, which we introduce here to describe our numerical results. We note that for random gaussian fermions an efficient algorithm has been proposed to compute the SRE [41], which could be in principle adapted to compute the Majorana spectrum. Here, we limit our analysis to small system sizes to perform a fair comparison with the $SYK_4$ model, which is the main focus of our work. Even with these small sizes, the agreement with the Laplace distribution for the Majorana spectrum of the $SYK_2$ model is excellent. Our results suggests therefore that the Laplace distribution is a generic feature of the Majorana spectrum for random Gaussian states. We note that obtaining this result analytically is far from trivial, since even if the state is Gaussian and one can use the Wick theorem to decompose the string of fermions, sampling is still required over a large number of contributions. Whether analytical progress is possible in this direction is an exciting question which is therefore left for future studies. Given the fact that the $SYK_2$ model is non-chaotic and integrable in many-body sense, our finding show that the Majorana spectrum encodes key and distinctive features of a quantum many-body system, such as its chaotic or integrable nature.

### 5.1.2 Stabilizer Rényi entropy for the ground state

In the previous section, we discussed the Majorana spectrum for the $SYK_2$ and SYK models. In this section, we focus on the Stabilizer Rényi Entropy (SRE) and the filtered SRE, which are related to the moments (Eq. 7) of the Majorana spectrum through Eq. 8. In Fig. 2, we present the Rényi index $\alpha$ dependence and system size $N$ scaling of the SRE $M_\alpha$ and the filtered SRE $\tilde{M}_\alpha$ for both $SYK_2$ and SYK models. We present all results averaged over disorder realizations of the Hamiltonian. For the averaging, we consider 800, 400, 200, 100, 25, and 5 disorder realizations for system sizes $N = 4, 6, 8, 10, 12, 14$, respectively. The disorder-averaged results are shown in Figure 2, with standard deviations indicated as error bars in all plots.

In Fig. 2 (a), $M_\alpha$ and $\tilde{M}_\alpha$ are shown as functions of the Rényi index $\alpha$ for both models with a system size of $N = 12$. We observe that the SRE and filtered SRE coincide for $\alpha = 1$, corresponding to the Shannon entropy. This behavior is expected, as the contributions from the identity operator and the parity operator, having square expectation values of *one* for pure fermionic state, yield *zero* entropy. For $\alpha \geq 2$, the filtered SRE becomes larger than the SRE. As the Rényi index $\alpha > 2$ increases, the contributions from the identity and parity operators become dominant, causing $M_\alpha$ for $\alpha \gtrsim 4$ to fail in capturing the qualitative differences in the Majorana spectrum $\Pi(x)$. Consequently, $M_\alpha$ with $\alpha \gtrsim 4$ becomes identical for both models. Interestingly, since the filtered SRE removes the contributions from the identity and parity operators, we find that $\tilde{M}_\alpha$ for $\alpha > 2$ is consistently larger for the SYK model compared to the $SYK_2$ model. This demonstrates that the filtered SRE effectively captures some characteristic features of the Majorana spectrum distribution.

In Fig. 2 (b), we compare the system size scaling $N$ of the SRE $M_2$ for the SYK and $SYK_2$ models. In both cases we observe that the SRE scales linearly with system size, albeit not reaching the maximal value of $N \log 2$. The degree of non-stabilizerness for the SYK model remains greater than that of the $SYK_2$ model for comparatively larger system sizes.

In Fig. 2 (c, d), we present the filtered SRE $\tilde{M}_\alpha$ for $\alpha = 2, 3, 4, 5$ as a function of $N$ for the $SYK_2$ and SYK models, respectively. Clearly, we observe that $\tilde{M}_\alpha$ also follows a linear scaling with system size for $\alpha \leq 5$. We note that for typical states it is expected that the filtered SRE to scale in the thermodynamic limit as $\tilde{M}_\alpha \sim D_\alpha N + c_\alpha$ with $D_\alpha = 1$ [34]. Our data for the SYK model shows a weak $\alpha$−dependence in the slope of $\tilde{M}_\alpha$ vs $N$, suggesting that larger system

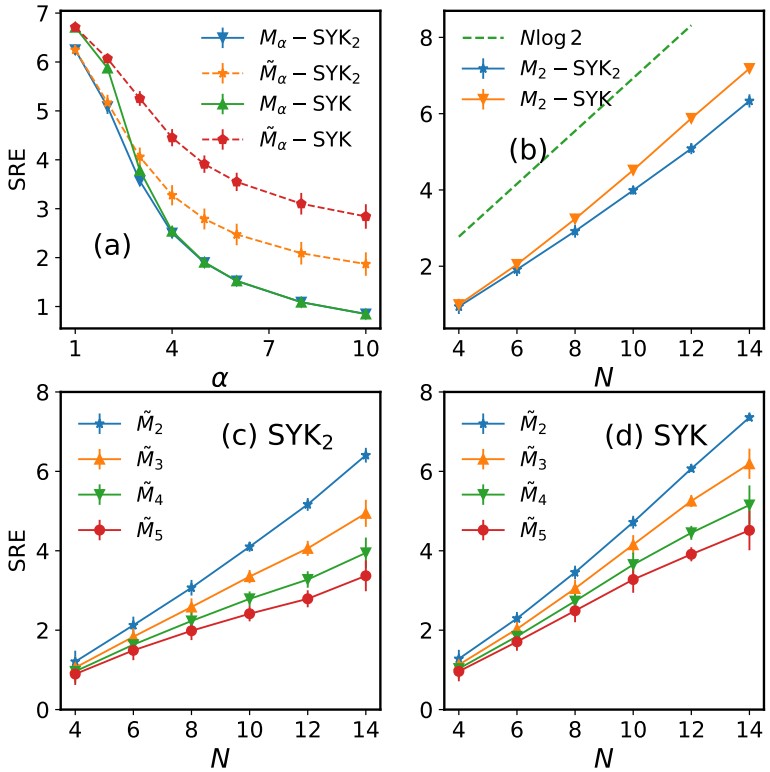

Figure 2: Stabilizer Rényi Entropy (SRE) for the ground state of the $SYK_2$ and SYK models: (a) The SRE $M_\alpha$ and the filtered SRE $\tilde{M}_\alpha$ are shown as functions of the Rényi index $\alpha$ for the $SYK_2$ and SYK models at system size $N = 12$. (b) The scaling of the SRE $M_2$ with system size $N$ is shown for both models and compared to the maximal entropy $N \log 2$. (c) The filtered SRE $\tilde{M}_\alpha$ for $\alpha = 2, 3, 4, 5$ is plotted as a function of $N$ for the $SYK_2$ model. (d) The filtered SRE $\tilde{M}_\alpha$ for $\alpha = 2, 3, 4, 5$ is plotted as a function of $N$ for the SYK model. All results are averaged over disorder realizations, with standard deviations shown as error bars.

sizes are needed to access the scaling regime.

## 5.2 Dynamics of non-stabilizerness under SYK and $SYK_2$ Hamiltonian

In this section, we discuss the time evolution of magic, or non-stabilizerness, starting from a stabilizer state. To investigate the dynamics of magic, we examine the quench dynamics of an initial product state under the evolution governed by the SYK (or $SYK_2$) Hamiltonian. The product states in the occupation basis are stabilizer states. Specifically, we consider an initial charge-density state in the occupation basis, represented as:

$$|\Psi_0\rangle = |1010\cdots\rangle. \tag{20}$$

At $t = 0$, the SYK (or $SYK_2$) Hamiltonian is switched on, evolving the state as:

$$|\Psi(t)\rangle = e^{-iHt}|\Psi_0\rangle. \tag{21}$$

The initial state has a fixed particle number $N_p = N/2$, and hence the state evolves within the half-filling sector of the Hamiltonian. We can express the time-evolved state in terms of energy

eigenvalues and eigenstates as follows:

$$|\Psi(t)\rangle = \sum_n c_n e^{-iE_n t}|E_n\rangle, \tag{22}$$

where $c_n = \langle E_n|\Psi_0\rangle$ are the amplitudes of the initial state in the energy eigenbasis. Once the time-evolved state at time $t$ is determined, we can compute the Majorana spectrum and the stabilizer Rényi entropy as functions of time. Below, we discuss how the Majorana spectrum and SRE evolve with time, starting from the initial product state, which is a stabilizer state.

### 5.2.1 Time evolution of Majorana spectrum

In Fig. 3 (a-e), we show the Majorana spectrum $\Pi(x)$ at different time instances, $t = 0.01, 0.5, 1.0, 2.0, 10.0$, under the evolution of the $SYK_2$ model starting from a product initial state. Similarly, the evolution of the Majorana spectrum at the same time instances under the SYK model is shown in Fig. 3 (f-j). The initial product state, being a stabilizer state, has exactly $2^N$ expectation values of $\langle\Psi_0|\hat{\mu}(\nu)|\Psi_0\rangle$ equal to $\pm 1$, with all others being *zero*. The spectrum is computed from system size $N = 10$ using ED for a single disorder realisation of the Hamiltonian.

In Fig. 3 (a, f), at a relatively short time, $t = 0.01$, the spectrum retains peaks at $x = \pm 1$ and $x = 0$. This distribution closely resembles the initial product state, with only slight broadening of the peak at $x = 0$. As time progresses, the distribution around $x = 0$ broadens significantly, as observed in Fig. 3 (b-d) and (g-i). Simultaneously, the height of the peaks at $x = \pm 1$ gradually decreases. As the state deviates further from the stabilizer product state, the number of expectation values $\langle\Psi(t)|\hat{\mu}(\nu)|\Psi(t)\rangle = \pm 1$ reduces from $2^N$.

At all time instances, we observe that the distribution is comparatively broader for the $SYK_2$ model compared to the SYK model. Additionally, the $SYK_2$ model's distribution gradually develops sharper peaks compared to the SYK model. At very long times, $t = 10$, the shape of the distribution under the evolution of the $SYK_2$ model closely resembles that of its ground state Majorana spectrum. This distribution fits well with a Laplace distribution. Similarly, under the evolution of the SYK model, the long-time distribution develops a dome shape, similar to the ground state Majorana spectrum of the SYK model, and fits well with a Gaussian distribution.

Under the time evolution governed by the $SYK_2$ and SYK Hamiltonian, the long-time state typically resides in a superposition of high-energy eigenstates of the $SYK_q$ Hamiltonian. This is because the initial energy density of the product state is far from the ground state energy density. Therefore, the similarity between the long-time Majorana spectrum and the ground state spectrum suggests that the high-energy eigenstates exhibit a Majorana spectrum similar to their corresponding ground states.

### 5.2.2 Time evolution of stabilizer Rényi entropy

In the previous section, we discussed the time evolution of the Majorana spectrum for a particular system size. Here, in Fig. 4, we show how the Stabilizer Rényi Entropy (SRE), averaged over disorder realizations, evolves with time for different system sizes. Specifically, we focus on $M_2$ for this analysis. As before, we compute $M_2[\Psi(t)]$ using exact diagonalization (ED) for system sizes up to $N = 8$, and for $N = 10, 12$, we employ Monte Carlo sampling.

In Fig. 4 (a, b), we present $M_2[\Psi(t)]$ as a function of time $t$ for the $SYK_2$ and SYK models, respectively, on a log-log scale. We observe that $M_2$ exhibits a power-law growth phase, $M_2 \sim t^x$ ($x > 0$), before saturating. For both models, $M_2$ saturates at a time scale $\tau_S$ which for the sizes explored in our simulation appear to be independent of system size, i.e. $\tau_S \sim \mathcal{O}(N^0)$. Further, we also checked that the entanglement entropy dynamics [65] after quench from

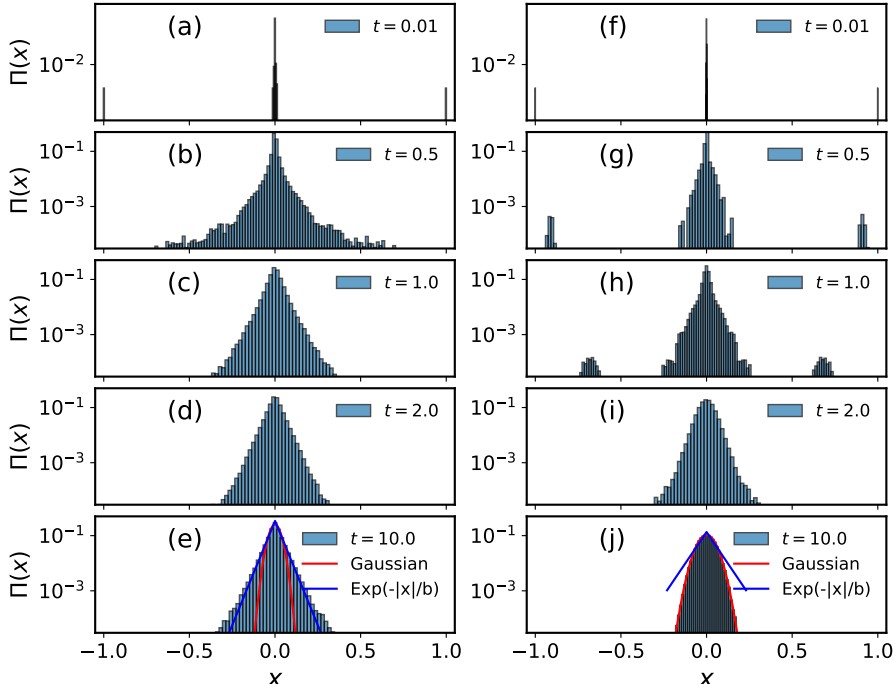

Figure 3: (a-e) Time evolution of the Majorana spectrum $\Pi(x)$ after a quench at $t = 0$ under $SYK_2$ is shown. The long-time state follows a Laplace distribution ($\sim \exp(-|x|/b)$), as illustrated in (e). (f-j) Time evolution of the Majorana spectrum $\Pi(x)$ after a quench at $t = 0$ under SYK is shown. The long-time state follows a Gaussian distribution, as illustrated in (j). The initial state is a product state given in Eq. 20. All the distributions are shown in semi-logarithmic ($\log y$) scale. This is shown for system size $N = 10$ using ED method.

product initial state behaves very similar to magic SRE dynamics and doesn't show system size dependence for system considered in this work. We mention that previous studies of the growth dynamics of SRE have found that SRE saturates at a time scale of $\mathcal{O}(\log N)$ for random circuits and a time scale of $\mathcal{O}(N)$ for chaotic Floquet many-body Hamiltonian [36]. Therefore, our all-to-all random coupling fermion model exhibits a different time scale for the dynamics SRE compared to that of the random circuits model. We expect this result to be related to the small system sizes considered here and that upon increasing the total size $N$ a more pronounced dependence of the saturating time with the size of the system should be visible.

In Fig. 4 (c), we compare the growth of $M_2$ for both models at system size $N = 8$ and find that the $SYK_2$ model surprisingly saturates relatively earlier than the SYK model, though both saturate at a time scale of $\mathcal{O}(1)$. Finally, in Fig. 4 (d), we plot the long-time saturation values of $M_2$ for both models as a function of system size $N$. We observe a linear scaling with system size, with the SYK model having a larger saturation value than the $SYK_2$ model, similar to what we found for the ground state of the corresponding models.

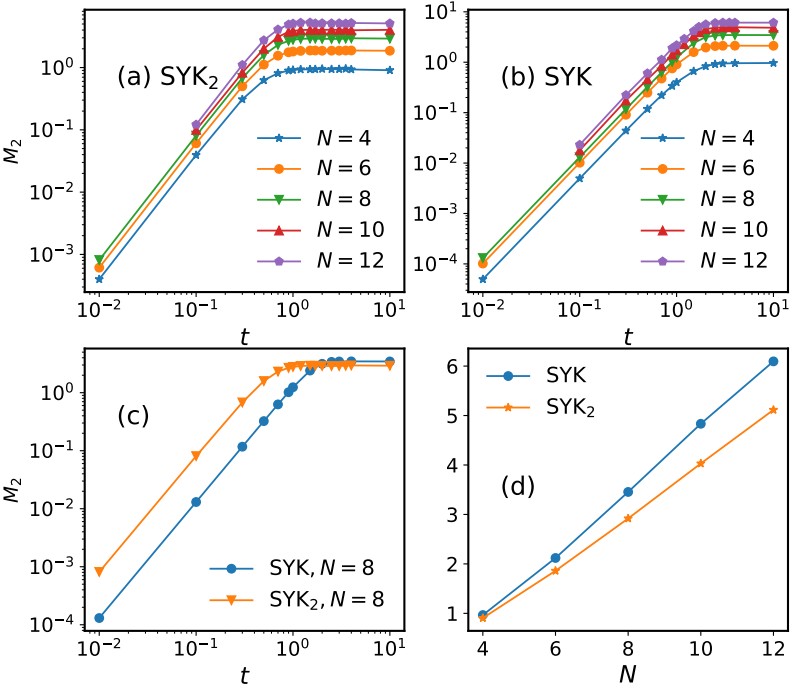

Figure 4: (a) Time evolution of SRE $M_2[\Psi(t)]$ under $SYK_2$ Hamiltonian is shown for different system sizes after quench at $t = 0$. (b) Time evolution of $M_2[\Psi(t)]$ under SYK Hamiltonian is shown for different system sizes starting after quench at $t = 0$. (c) Comparision of time evolution of SRE $M_2[\Psi(t)]$ under $SYK_2$ and SYK Hamiltonian for $N = 8$ system. (d) The SRE ($M_2[\Psi(t \gg 1)]$) of long-time states ($t \gg 1$) (saturation values) are plotted as a function of system sizes $N$ for both $SYK_2$ and SYK model. The initial state is a product state as given in eqn.20. The plot in (a-c) are shown in semi-logarithmic ($\log y$) scale and the plot in (d) are shown in linear scale.

# 6 Conclusions

In this work we have studied the Sachdev-Ye-Kitaev model of complex fermions with all-to-all random interactions from the point of view of its resource theory content, specifically its non-stabilizerness. Using exact diagonalizaton and a Monte Carlo sampling in the space of Majorana strings we have computed the Majorana spectrum of the SYK model and its associated stabilizer Renyi entropy. We have presented numerical evidence for the Majorana spectrum to display a Gaussian distribution, in accordance with known results on chaotic many-body systems. We have compared this result with the Majorana spectrum of the $SYK_2$ model describing random Gaussian fermions, which we have shown to display an exponential distribution. An interesting open question concerns whether this result can be obtained analytically using the properties of fermionic Gaussian states [41].

For what concerns the SRE in the ground-state, and in particular its filtered version that we have used throughout this work, we have shown how this quantity is generically larger for the SYK model as compared to the free fermion case described by the $SYK_2$. None of these two cases however saturates the maximum upper bound for SRE. Finally, we have discussed the real-time dynamics and spreading of magic and of the Majorana spectrum in the two models.

Our results show that the the second-Renyi entropy spreads as a power-law in time and that it saturates on a time scale which depends weakly on system sizes, at least for the values explored in this work. Whether this is a genuine feature of the fully connected nature of the SYK model or rather due to strong finite size effects is an open question for the future.

Future directions include the investigation of how the non-stabilizerness of the SYK model can be estimated for larger system sizes, for example using clever representations of the wave function [66–69] or the large $N$ limit and field theory and replica techniques, particularly given our results on the self-averaging nature of the Majorana spectrum. In addition, an interesting future direction could be to explore the role of quantum measurements on the dynamics of magic in the SYK model and possible phase transitions in the quantum information resource content of the theory [70–72].

*Note Added:* Upon completion of this work, we became aware of Refs. [73,74], which also have partial overlap with our study of the non-stabilizerness of the Sachdev-Ye-Kitaev model. We briefly discuss the extent of this overlap below. Ref. [73] investigates the unitary dynamics of the stabilizer Rényi entropy (SRE) for the SYK model in the spin representation with total $S^z = 0$, which directly corresponds to our complex SYK model at half-filling and shows very similar steady-state values. Ref. [74] examines the ground-state SRE of the Majorana SYK model, which can be compared to our results for the complex SYK model; their results indicate a slightly lower SRE for the Majorana case, though both are of comparable magnitude.

## Acknowledgments

We would like to thank Xhek Turkeshi for valuable comments on our manuscript, in particular regarding the Majorana spectrum and for pointing Ref. [75] to our attention.

**Funding information** This work has received funding from the European Research Council (ERC) under the European Union's Horizon 2020 research and innovation programme (Grant agreement No. 101002955 – CONQUER) The computations were performed on the Collége de France IPH computer cluster.

## A  Additional results on ground state Majorana spectrum

In the main text (Fig. 1), we have shown the Majorana spectrum for the ground state of the SYK and SYK$_2$ models for a particular quenched disorder realization of the Hamiltonian. Here, we provide additional results on the Majorana spectrum for the ground state of both models for other disorder realizations of the Hamiltonian. Furthermore, we discuss the Majorana spectrum for an ensemble of different disorder realizations across various system sizes for both models, computed using exact diagonalization (ED).

In Fig. 5 (a-e), we present the Majorana spectrum $\Pi(x)$ for five randomly chosen disorder realizations of the SYK$_2$ Hamiltonian for system size $N = 8$, shown on a semi-logarithmic scale. Similarly, in Fig. 5 (f-j), we present $\Pi(x)$ for five different disorder realizations of the SYK model. As before, we observe that $\Pi(x)$ for each disorder realization of the non-chaotic SYK$_2$ model fits a Laplace distribution (blue fitting curve), whereas for the chaotic SYK model, it fits a Gaussian distribution (red fitting curve). Additionally, we notice that the spectrum exhibits weak sample-to-sample fluctuations.

In Fig. 6, we show the Majorana spectrum $\Pi(x)$ of an ensemble constructed from the spectra of 10 different disorder realizations for $N \leq 8$ and 3 different disorder realizations for $N = 10$. In Fig. 6(a-c), we present the spectrum $\Pi(x)$ of the ensemble for the SYK$_2$ model for

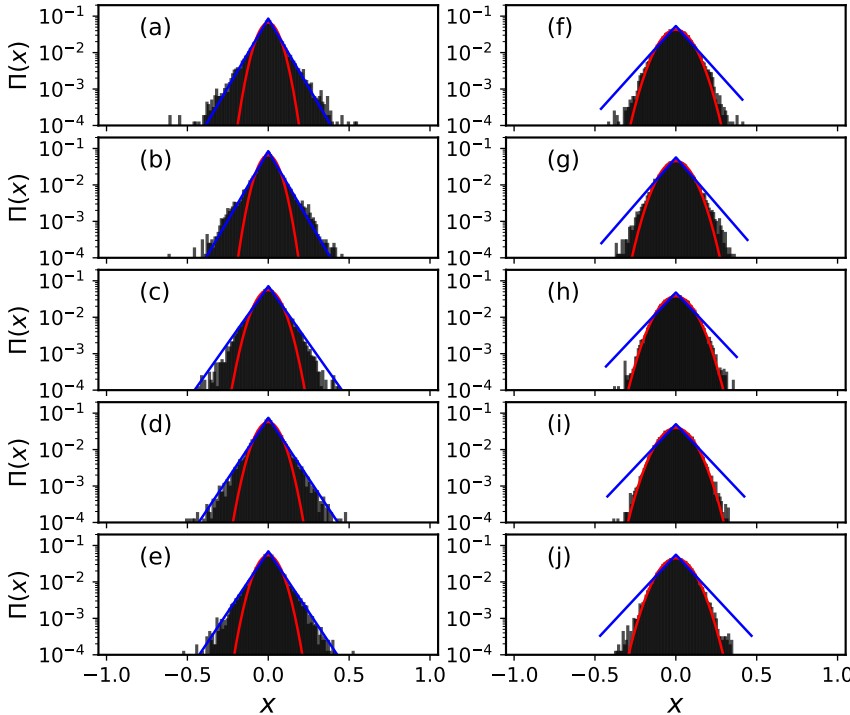

Figure 5: (a-e) The Majorana spectrum $\Pi(x)$ for five randomly chosen disorder realizations of the $SYK_2$ model are shown here. (f-j) Similarly, $\Pi(x)$ for five randomly chosen disorder realizations of the SYK model are presented. The Gaussian (red) and exponential Laplace (blue) fittings are also shown in the histograms, indicating that each realisation of SYK model fits a Gaussian distribution, while the $SYK_2$ model fits a Laplace distribution. The spectrum is computed from system size $N = 8$.

system sizes $N = 4, 6, 8, 10$, respectively. Similarly, in Fig. 6(d-f), we show the results for the ensemble of the SYK model.

We observe that for very small system size $N = 4$, the distribution is broad and almost uniform for both models. For $N > 4$, we find that $\Pi(x)$ for the ensemble behaves similarly to individual disorder realizations, namely, exhibiting a sharp peak and broad-tail Laplace distribution for the $SYK_2$ model and a Gaussian distribution for the chaotic SYK model. As the system size increases, the distribution becomes significantly narrower. Furthermore, the distribution for the SYK model is narrower than that of the $SYK_2$ model, as captured by the stabilizer Rényi entropy (SRE) discussed in the main text.

## B   Monte Carlo update algorithm

For updating the Majorana string, we consider a combination of two site updates such that the total number of Majorana operators in the string remains even. This is because odd parity Majorana string operators give zero expectation values due to the parity superselection rule for fermions. We can view the $2N$-length Majorana string operator as being constructed from

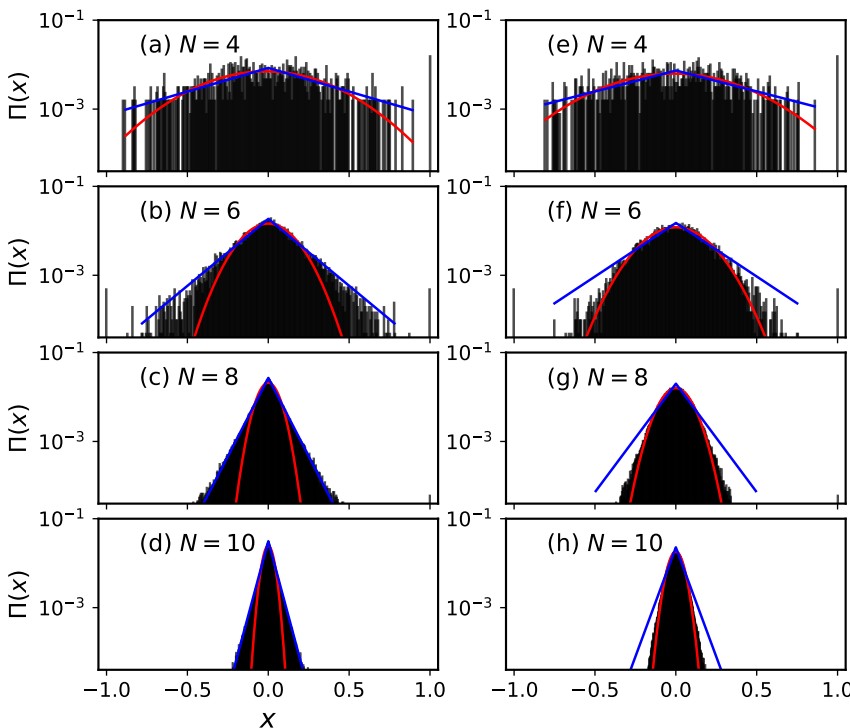

Figure 6: (a-c) The Majorana spectrum $\Pi(x)$ for an ensemble constructed from the Majorana spectra of different disorder realizations of the $\text{SYK}_2$ model is shown for system sizes $N = 4, 6, 8, 10$ respectively. (d-f) Similarly, the Majorana spectrum for an ensemble constructed from different disorder realizations of the SYK model is shown for $N = 4, 6, 8, 10$. The Gaussian (red) and exponential Laplace (blue) fittings are also shown in the histograms showing SYK-ensemble follow Gaussian and $\text{SYK}_2$ follow Laplace distribution for system sizes $N > 4$.

$N$ sites of complex fermions, with each site spanned by the following four operators:

$$\text{OP}_i = \left\{ \eta_i, \chi_i, \eta_i \chi_i, \mathcal{I} \right\}. \tag{B.1}$$

We provide the two-site update procedure in Algorithm 2. We start the update procedure using the low-weight Majorana string operator, where most of the Majorana operators in the string are absent.

## C  Benchmarking the Monte Carlo sampling procedure

### C.1  Convergence for filtered SRE

In this section, we benchmark and compare the Monte Carlo sampling method with the exact diagonalisation method for computing the filtered SRE for both models. Recall that in computing the filtered SRE, we set the probabilities $\sigma_v[\mathcal{I}, \mathcal{P}] = 0$ for the identity $\mathcal{I}$ and the parity operator $\mathcal{P}$, as discussed in the main text.

---

**Algorithm 2** Majorana String Update Algorithm

---

 1: **function** UPDATE(Input Majorana string operator)
 2:     **while** true **do**
 3:        Select two distinct sites $i, j$ randomly.
 4:        Save the original operators for these two sites.
 5:        Propose two new operators randomly chosen from the lists $\text{OP}_i$ and $\text{OP}_j$.
 6:        Apply the changes with two new operator in Majorana string.
 7:        **if** the proposed string is even parity **then**
 8:           Compute the $\text{i}^{\nu^T \omega_L \nu}$ factor.
 9:           **return** new Hermitian Majorana string operator.
10:        **else**
11:           Restore the original operators.
12:        **end if**
13:     **end while**
14: **end function**

---

In Figure 7, we present the filtered SRE computed using Monte Carlo sampling, as described in Tables 1 and 2, alongside results obtained from exact diagonalisation up to system size $N = 12$. We show Monte Carlo results for different sample sizes $N_S$, as indicated in the legend, for the SYK model in Figure 7(a) and the $\text{SYK}_2$ model in Figure 7(b). These results are computed for the ground state at half-filling, $N_p = N/2$. We find that the results for different $N_S \geq 10^5$ are nearly indistinguishable, and the result for $N_S = 5 \times 10^5$ matches the exact diagonalisation (ED) result. This benchmarks the Monte Carlo algorithm and demonstrates excellent convergence of the Monte Carlo results with sample size. As we discuss next, direct sampling of the SRE, however, requires a larger sample size and shows relatively poorer convergence compared to the filtered SRE.

After obtaining the filtered SRE data, the full SRE result can be reconstructed by manually adding the contributions of the identity and parity operators, whose expectation values are known: $\langle\Psi|\mathcal{I}|\Psi\rangle^2 = 1 = \langle\Psi|\mathcal{P}|\Psi\rangle^2$. We have verified this procedure for smaller system sizes using ED and sampling data. All results presented in the main text for the ground state SRE and filtered SRE are computed using sampling of the filtered SRE. We consider a maximum sample size of $N_S = 5 \times 10^5$ for systems with $N = 10, 12, 14$.

## C.2   Convergence for direct sampling of SRE

In this section, we present a benchmark analysis of the Monte Carlo sampling procedure for the SRE of the ground state for system sizes up to $N = 12$, using a single disorder realization of the $\text{SYK}_2$ model. We also examine the convergence of the SRE results as a function of the sampling size $N_S$.

Figure 8 shows the SRE $M_\alpha$ ($\alpha = 1, 2$) of the ground state at $\mu = 0$ without fixing total particle number of the $\text{SYK}_2$ model as a function of system size for different sampling sizes: $N_S = 10^4, 10^5, 10^6$. The data exhibits good convergence even for relatively small sampling sizes when $N \leq 10$. For $N = 12$, the results obtained with $N_S = 10^5$ and $N_S = 10^6$ are close, though convergence is relatively poorer compared to the filtered SRE results discussed above. Additionally, we include exact diagonalisation (ED) results for system sizes up to $N = 12$, as indicated in the legend.

Based on this benchmark, we adopt a sampling size of $N_S = 5 \times 10^5$ for system sizes $N = 10$ and $N = 12$ in the results presented in the main text for the time evolution of an initial state.

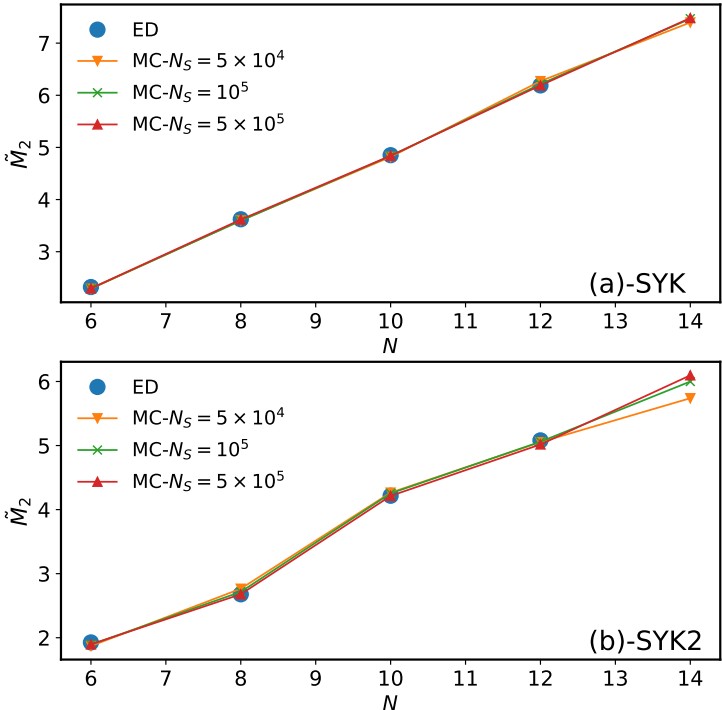

Figure 7: The convergence of filtered SRE $\tilde{M}_2$ with sampling size $N_S$ is shown for different system sizes up to $N = 14$ for both SYK (a) and SYK$_2$ (b) models. The exact diagonalization (ED) results obtained for system sizes up to $N = 12$ match exactly with the Monte Carlo sampling results for $N_S = 5 \times 10^5$.

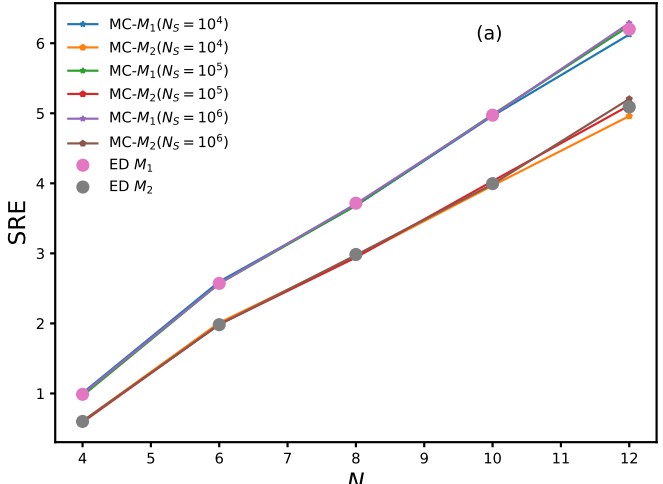

Figure 8: The convergence of SRE $(M_1, M_2)$ with sampling size $N_S$ is shown for different system sizes up to $N = 12$. The exact diagonalization (ED) results obtained for system sizes up to $N = 12$ match closely with the Monte Carlo sampling results.

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
