# Peer review of "Non-Stabilizerness of Sachdev-Ye-Kitaev Model"

_SciPost Physics, doi:SciPost Phys. 19, 159 (2025)_

## Round 2 · Referee Report · Pengfei Zhang (Referee 1) · 2025-8-30

Report

Quantum entanglement and magic are two distinct probes of a quantum system’s capacity for quantum information processing. In recent years, there have been various breakthroughs in understanding the entanglement properties of many-body quantum systems, while the study of magic has only recently begun to attract attention. In this manuscript, authors study the magic by considering the Stabilizer Renyi Entropy in both the chaotic SYK4 model and the SYK2 model that describes random hopping Majorana fermions. The main observation is the qualitatively different distribution for the Majorana spectrum: SYK4 shows a Gaussian spectrum, as proposed in previous works while SYK2 exhibits an exponential Laplace distribution.

I believe these results are of particular interests to the field of quantum dynamics and SYK-like models: They provide valuable examples for the behavior of Majorana spectrum/stablizer Renyi entropy in concrete many-body systems. In addition, given the close relationship between SYK models and gravity, this may inspire the study of stablizer Renyi entropy in holography. Therefore, I'm happy to suggest the publication of this work in SciPost Physics.

My minor comments to considered by authors include:

  1. The Majorana spectrum is auctually the expectation of a Majorana string. For the SYK2 model, the expectation should satisfies the Wick's theorem, which relates a generic Majorana spectrum to two-point correlators. Does this directly lead to the observed exponential Laplace distribution directly?

  2. In the main discussion, authors consider the complex SYK model. Could authors comment on this choice? In particular, is there any difference when considering the Majorana SYK model? Does charge conservation play any role?

Recommendation

Publish (meets expectations and criteria for this Journal)

  • validity: high
  • significance: good
  • originality: good
  • clarity: high
  • formatting: excellent
  • grammar: excellent

Author:  Surajit Bera  on 2025-11-12  [id 6016]

(in reply to Report 1 by Pengfei Zhang on 2025-08-30)
Category:
answer to question

We thank the referee for recommending our article for publication. We have attached a PDF document containing our responses to the minor comments raised by the referee.

Attachment:

Referee_response_Magic_Referee1.pdf

Pengfei Zhang  on 2025-11-12  [id 6020]

(in reply to Surajit Bera on 2025-11-12 [id 6016])

I thank the authors for their detailed explanation on my minor comments. As I mentioned in the report, I believe the work is of particular interests to the field. Therefore, I suggest its publication in its current form.

---

## Round 2 · Referee Report · Anonymous (Referee 2) · 2025-9-19

Strengths

  1. Timely topic
  2. Potential interest in different communities

Weaknesses

  1. Results are only presented for relatively small sizes even for SYK2 which is non-interacting.

Report

The authors study the Stabilizer Renyi Entropy and the distribution of the Majorana spectrum, measurements of magic, for both the SYK2 and the SYK4 model for complex fermions. Both have random couplings though the former is non-interacting and the latter is strongly interacting. The main results are that the SRE has the same linear scaling with N in both models though in the interacting one is much larger. The distribution is different in both cases, exponential in the non-interacting model while it is a Gaussian in the interacting case.

I think this is an interesting addition to the literature because these results potentially provide a way to characterize the impact of interaction on magic. Moreover, these results could also be of interest in the low dimensional quantum gravity community due to the duality of the interacting SYK model to JT gravity. Moreover, this community is currently interested in quantum information concepts and idea.

I am glad to accept the paper for publication after address the following comments: 1. I am surprised that computationally the SYK_2 does not allow a short-cut to reach much larger sizes. Even numerically, the application of the Wick theorem should make possible to reach much larger sizes. Could the author comment on this? 2. I would also ask the authors to provide a more detailed description of similar calculations in the recent literature. They mention overlap with 72,73. They should elaborate. More specifically, they should comment to what extent their results are universal at least in the context of Fermionic theories. Is expected a linear scaling with N of the SRE in all fermionic systems? Is a Gaussian and an exponential distribution generic for interacting and integrable fermions? 3. Why do the author study the SYK with Dirac fermions instead of the flavor with Majoranas? I would expect that the Majorana one is more interesting because larger sizes are available and because it is more directly related to Pauli spins. 4. The SYK references require some tweaking. When referring to SYK the first time, [45] is irrelevant and [49] should be cited and Kitaev talk should likely be first. SYK models for Dirac fermions in the context of RMT were studied much earlier than [43] but I do not want to be too intrusive with respect to citations. When referring to black holes, the authors should refer to seminal papers and not to what look like a review. The relevance to gravity/black holes is already in Kitaev talk and also in [49]. There were several early papers that pointed out this relation. For instance, https://arxiv.org/abs/1611.04650 https://arxiv.org/abs/1610.03816 https://link.springer.com/article/10.1007/JHEP08(2017)136 https://journals.aps.org/prx/abstract/10.1103/PhysRevX.5.041025 The authors could take a look a choose a few.

Requested changes

I ask the authors to modify the manuscript according to the question raised in the report

Recommendation

Ask for minor revision

  • validity: high
  • significance: good
  • originality: good
  • clarity: top
  • formatting: excellent
  • grammar: excellent

Author:  Surajit Bera  on 2025-11-12  [id 6017]

(in reply to Report 2 on 2025-09-19)
Category:
answer to question

We thank the referee for their valuable questions and comments. We hope that the referee will be satisfied with our responses and the minor revisions made to the manuscript and will recommend it in Scipost Physics for publication.

Attachment:

Referee_response_Magic_Referee2.pdf

---

## Round 2 · Referee Report · Anonymous (Referee 3) · 2025-10-13

Strengths

  • Timely topic
  • Accurate presentation of the work and of the results
  • Scientifically sound

Weaknesses

  • It is a purely numerical work and does not attempt to improve me methodology, nor to interpret (analytically) the results

Report

This manuscript provides a very detailed analysis of the non-stabilizerness for two prototypical models of random all-to-all interactions: the full SYK and its quadratic version. The former is known to be more complex and interesting from many points of views, compared to the second and quantum complexity, measured by non-stabilizerness, confirms this expectation. Beside the quantitative bechmarks for the different behavior contained in this work, the only new result seems to be the observation that the Mayorana spectrum for the quadratic model follows a Laplace distribution, thus decaying slower than the Gaussian behavior of the full SYK case. However, this result remains just an empirical observation and no attempt is made to understand it, while it seems rather natural that such behavior should stem from the decomposibility of correlations functions into 2-point functions. But the manuscript does not dwelve in such analysis.
Thus, while I fully support the publication of this work, I do not think it meets the high standard of innovation and relevance required by the flagship journal and I recommend it acceptance in Scipost Core.

Requested changes

  • On page 4, the sentence " On the other hand, odd-parity strings can describe as logical operations." should be corrected for English;
  • After eq. (12): can the author comment if they check that another filling fraction produces similar results or has $N_p=N/2$ been choosen for a particular reason to be clearly stated?
  • Eqs. (17) and (18) contains delta-function contributions not visible in the plots of Figs . 1: can the authors explain why?
  • Does the choice of the initial state in eq. (20) matter for the evolution or other separable, stabilizer states behave similarly?
  • I think that a little more details should be given in the conclusion on the relation between this work and those in ref. [72,73]

Recommendation

Accept in alternative Journal (see Report)

  • validity: good
  • significance: good
  • originality: good
  • clarity: top
  • formatting: perfect
  • grammar: excellent

Author:  Surajit Bera  on 2025-11-12  [id 6018]

(in reply to Report 3 on 2025-10-13)
Category:
answer to question
reply to objection

We thank the referee for reviewing our manuscript. However, we respectfully disagree with the referee’s comments. A detailed response addressing each point is provided in the attached PDF.

Attachment:

Referee_response_Magic_Referee3.pdf

---

## Round 3 · Referee Report · Anonymous (Referee 2) · 2025-11-14

Report
Recommendation
Publish (meets expectations and criteria for this Journal)

---

## Round 3 · Author Response

List of changes
(1) Based on referee 2's comments 1 and 2, we have added the following text on page 10 :
''We note that for random gaussian fermions an efficient algorithm has been proposed to compute
the SRE [41], which could be in principle adapted to compute the Majorana spectrum.
Here, we limit our analysis to small system sizes to perform a fair comparison with the
SYK4 model, which is the main focus of our work. Even with these small sizes, the
agreement with the Laplace distribution for the Majorana spectrum of the SYK2 model is
excellent. Our results suggests therefore that the Laplace distribution is a generic feature
of the Majorana spectrum for random Gaussian states. We note that obtaining this result
analytically is far from trivial, since even if the state is Gaussian and one can use the Wick
theorem to decompose the string of fermions, sampling is still required over a large number
of contributions. Whether analytical progress is possible in this direction is an exciting
question which is therefore left for future studies.''
(2) According to referee 2's suggestions (comment 2) and referee 3's requested change, we elaborate briefly on the overlap with ref 73 and ref 74 in the "Note added" section. The added text reads as below:
''Upon completion of this work, we became aware of Refs. [73,74], which also have partial overlap with our study of the non-stabilizerness of the Sachdev–Ye–Kitaev model. We briefly discuss the extent of this overlap below. Ref. [73] investigates the unitary dynamics of the stabilizer Rényi entropy (SRE) for the SYK model in the spin representation with total Sz = 0, which directly corresponds to our complex SYK model at half-filling and shows very similar steady-state values. Ref. [74] examines the ground- state SRE of the Majorana SYK model, which can be compared to our results for the complex SYK model; their results indicate a slightly lower SRE for the Majorana case, though both are of comparable magnitude''.
(3) According to referee 2's suggestion (comment 4), we have now incorporated the correct reference for SYK and related literature on page 2-3.
(4) According to referee 3's requested change, we have corrected the sentence on page 4, which reads now : 'On the other hand, odd-parity strings can act as logical operations' .

---

## Round 3 · List of Changes

(1) Based on referee 2's comments 1 and 2, we have added the following text on page 10 :
''We note that for random gaussian fermions an efficient algorithm has been proposed to compute
the SRE [41], which could be in principle adapted to compute the Majorana spectrum.
Here, we limit our analysis to small system sizes to perform a fair comparison with the
SYK4 model, which is the main focus of our work. Even with these small sizes, the
agreement with the Laplace distribution for the Majorana spectrum of the SYK2 model is
excellent. Our results suggests therefore that the Laplace distribution is a generic feature
of the Majorana spectrum for random Gaussian states. We note that obtaining this result
analytically is far from trivial, since even if the state is Gaussian and one can use the Wick
theorem to decompose the string of fermions, sampling is still required over a large number
of contributions. Whether analytical progress is possible in this direction is an exciting
question which is therefore left for future studies.''
(2) According to referee 2's suggestions (comment 2) and referee 3's requested change, we elaborate briefly on the overlap with ref 73 and ref 74 in the "Note added" section. The added text reads as below:
''Upon completion of this work, we became aware of Refs. [73,74], which also have partial overlap with our study of the non-stabilizerness of the Sachdev–Ye–Kitaev model. We briefly discuss the extent of this overlap below. Ref. [73] investigates the unitary dynamics of the stabilizer Rényi entropy (SRE) for the SYK model in the spin representation with total Sz = 0, which directly corresponds to our complex SYK model at half-filling and shows very similar steady-state values. Ref. [74] examines the ground- state SRE of the Majorana SYK model, which can be compared to our results for the complex SYK model; their results indicate a slightly lower SRE for the Majorana case, though both are of comparable magnitude''.
(3) According to referee 2's suggestion (comment 4), we have now incorporated the correct reference for SYK and related literature on page 2-3.
(4) According to referee 3's requested change, we have corrected the sentence on page 4, which reads now : 'On the other hand, odd-parity strings can act as logical operations' .

---

## Editorial Decision

published